# Is the Complement System Dysregulated in Preeclampsia Comorbid with HIV Infection?

**DOI:** 10.3390/ijms25116232

**Published:** 2024-06-05

**Authors:** Sumeshree Govender, Mikyle David, Thajasvarie Naicker

**Affiliations:** Optics and Imaging Centre, Doris Duke Medical Research Institute, College of Health Sciences, University of KwaZulu-Natal, Durban 4000, South Africa; sumi123gov@gmail.com (S.G.); mikyledavid101@gmail.com (M.D.)

**Keywords:** complement, HIV, innate, immunity, preeclampsia, pregnancy

## Abstract

South Africa is the epicentre of the global HIV pandemic, with 13.9% of its population infected. Preeclampsia (PE), a hypertensive disorder of pregnancy, is often comorbid with HIV infection, leading to multi-organ dysfunction and convulsions. The exact pathophysiology of preeclampsia is triggered by an altered maternal immune response or defective development of maternal tolerance to the semi-allogenic foetus via the complement system. The complement system plays a vital role in the innate immune system, generating inflammation, mediating the clearance of microbes and injured tissue materials, and a mediator of adaptive immunity. Moreover, the complement system has a dual effect, of protecting the host against HIV infection and enhancing HIV infectivity. An upregulation of regulatory proteins has been implicated as an adaptive phenomenon in response to elevated complement-mediated cell lysis in HIV infection, further aggravated by preeclamptic complement activation. In light of the high prevalence of HIV infection and preeclampsia in South Africa, this review discusses the association of complement proteins and their role in the synergy of HIV infection and preeclampsia in South Africa. It aims to identify women at elevated risk, leading to early diagnosis and better management with targeted drug therapy, thereby improving the understanding of immunological dysregulation.

## 1. Introduction

In low- and middle-income countries, maternal mortality and morbidity are public health issues with a lack of precise data with regard to their prevalence and aetiology [1]. Hypertensive disorders of pregnancy (HDP) complicate up to 10% of pregnancies [2]. Hypertensive disorders of pregnancy may be classified into four groups: (1) chronic hypertension, (2) gestational hypertension that arises, (3) preeclampsia (PE)/eclampsia, and (4) superimposed PE on chronic hypertension [3]. Furthermore, one of the biggest health concerns in the world today is Human immunodeficiency virus (HIV) infection. 

The most current estimate of HIV cases worldwide in 2023 was 39 million [4]. In South Africa, non-pregnancy-related infections (mainly HIV infection) account for 2.6 million maternal deaths [4], while hypertensive disorders of pregnancy (mainly PE) constitute 88 deaths per 100,000 live births [5]. Preeclampsia is a multisystem, idiopathic condition that affects pregnant women and is characterized by the development of proteinuria and hypertension in a previously normotensive woman during midgestation [6,7]. It is one of the leading causes of maternal and foetal morbidity, with the global prevalence reported to be 2–4%, with approximately 46,000 maternal deaths and approximately 500,000 foetal and newborn deaths occurring annually [1].

The pathophysiology of PE is not yet elucidated; however, there is a deficient development of maternal tolerance to the foetus or an altered maternal immune response due to excessive activation of neutrophils and monocytes [8]. Notably, monocytes synthesise considerable amounts of pro-inflammatory chemokines and cytokines [9]. Additionally, CD8+, CD4+ and dendritic cells also escalated the pro-inflammatory response. The pro-inflammatory cytokines increase vascular permeability and induce trophoblast cell apoptosis. Furthermore, they also activate and damage endothelial cells, exacerbating the exaggerated inflammatory response [10].

The synergy of HIV infection and PE comes from antagonistic immune responses, as PE is associated with an exaggerated immune response, whilst HIV infection dampens the immune response [11]. The prevalence of PE and HIV is reported to be 15–19% in Sub-Sharan Africa [12]. Notwithstanding, adaptive and innate immunity are connected through the complement system [13]. The complement system forms part of our innate immune response and functions to opsonize target surfaces, induce pro-inflammatory responses, and lyse cells and pathogens. Furthermore, it defends the host by removing apoptotic cells, damaged tissue, and immune complexes, ensuring homeostasis maintenance [14]. 

In PE, C1q, C3a, C3b, C5a, and C5b-9 complement components are upregulated, suggesting an increase in complement activation that leads to opsonization, which increases the HIV viral load [15]. Subsequently, these complement components cause the release of excess inflammatory cytokines by pro-inflammatory T cells and a decrease in anti-inflammatory and regulatory cytokines [14]. Thus, PE and HIV infection create an imbalance, leading to chronic immune activation via excessive or dysregulated complement activation. Excessive or dysregulated complement activation can lead to inflammatory pathology [10]. This review helps to elucidate the role of the complement system in the duality of HIV infection and PE. 

## 2. The Complement System 

The complement system consists of more than 30 proteins present in the plasma and on cell surfaces [16]. The activation of immune cells when activated leads to the opsonization, inflammation, and lysis of potential pathogens, immune complexes, and apoptotic cells [17]. Three different routes initiate complement activation: the classical pathway (CP), the lectin pathway (LP), and the alternative pathway (AP), which combine to form a single common pathway [18,19]. Whilst the vital component C3 is activated by all three pathways the nature of recognition determines the fate of the opsonin [13]. 

### 2.1. Complement Activation 

When the complement system is activated, structural changes, proteolytic cleavages, and the formation of proteolytic and lytic complexes occur. 

The CP is triggered upon a humoral response. Apoptotic cells, damaged cells that have exposed cellular structures, and some microbes all experience direct C1q deposition. It is also targeted by pattern recognition molecules including Immunoglobulin G (IgG), and Immunoglobulin M (IgM) [19]. When the C1 complex adheres to the target cell, C4, C2, and the C3/C5 convertase are active, enabling the activation of the CP. Thereafter C3 is activated at the terminal pathway, leading to the formation of the C5b-9 complex, the membrane attack complex (MAC), at the cell membrane [20]. 

The LP is triggered when lectin proteins, mannose-binding lectin (MBL), or one of the ficolin’s (ficolin’s 1, 2, and 3) recognize and bind pathogen-associated molecular patterns (PAMPs) [21,22]. With the MBL-associated serine protease (MASP) enzymes, MASP1, MASP2, and MASP3, the lectins form complement-activating complexes. Thereafter, the LP functions similarly to those of the CP in activating components C2 and C4 that lead to the formation of C3 convertase, C4b2a [22].

Due to C3′s spontaneous activation of appropriate acceptor sites, the AP differs from both the CP and LP in that it is constantly “turned on” [23]. By hydrolysing the internal C3 thioester bond, the AP is naturally gradually activated [24]. It then undergoes further spontaneous activation and versatility in binding to a variety of different proteins, lipids, and carbohydrate structures on microbes and other foreign surfaces [25]. The thioester bond in C3 hydrolyses spontaneously to generate C3(H2O) (C3 convertase), which has a different structure and can interact with plasma protein factor B. After factor B is bound by C3(H2O), factor D, a plasma protease, cleaves factor B into Ba and Bb. The latter two components then combine to produce the C3(H2O) Bb complex [26]. Thereafter, Factor D cleaves Factor B to generate the C3 convertase C3bBb. This complex is stabilized by the presence of plasma properdin, and Factor B is subsequently recruited to the bound C3b complex [27]. 

C3 convertases, which split C3 into its active components, C3a and C3b, are produced by the convergence of all three routes [28]. In addition to assisting in activating a C5 convertase, C3b forms a covalent bond with the activating surface of its target. The generated layer of C3b molecules opsonizes the target and promotes its phagocytosis [19]. C5 is split into C5a and C5b by C5 convertase. Notably, C3a and C5a are known as anaphylatoxins which play a significant role in inflammation and acute allergic reactions. Lastly, the MAC (C5b-9) is created when C5b joins forces with C6, C7, C8, and C9 to trigger lysis (Figure 1) [29,30]. 

### 2.2. Complement Regulators

Regulation of the complement system is mediated by specific molecules known as complement regulators. They function by blocking the activation or initiation of their destruction, playing a crucial role in preventing damage to healthy cells. Additionally, strict complement regulation is necessary to stop an attack on both the foetus and self-tissue [31]. However, to detect and eliminate damaged and dying cells and to enable strong activation against invasive microbes, an ideal level of ongoing small grade complement activation is maintained.

The initiation, amplification, and production of effectors such as opsonin, MAC, and proinflammatory anaphylatoxins occur at levels in which complement activation may be controlled. Complement regulators are molecules that are soluble or membrane-bound and can prevent complement activation; examples of soluble inhibitors are C1 inhibitor (C1INH), C4b binding protein (C4BP), factor H (FH), clusterin, and vitronectin [3]. 

C1 inhibitor specifically inhibits initiation of the CP and LP by preventing C1r, C1s, and MASP activation. C3 inhibitor factor H is the primary soluble regulator of the AP [32]. Since C3 is an essential component in three different complement initiation pathways, the convertases are the target of several inhibitors. FH breaks down the alternative pathway C3 convertase (C3bBb). Similar roles for C4BP include facilitating C4b’s inactivation and managing C4b2a, the CP C3 convertase [3].

Additionally, by enhancing Factor I-mediated degradation of the C3 convertase, C3b binding of CR1 (CD35) increases the amount of C3b accessible for downstream complement activation following inflammation [3].

The only positive complement system regulator is properdin, which functions in part against FH by maintaining the stability of the AP C3-convertase, C3bBb [33]. Furthermore, it has been demonstrated that autologous complement activation is inhibited by three membrane proteins, which are found on the surface of all cell types, degradation-accelerating factor (DAF; CD55), membrane cofactor protein (MCP; CD46), and membrane inhibitor of reactive lysis (CD59) that work to protect self-cells from complement-mediated damage [34]. By hastening the breakdown of the C3 and C5 convertases, CD55 renders them inactive. The serum protease factor cleaves cell-bound C4b and C3b in the presence of the cofactor, CD46. CD59 controls the synthesis of MAC by limiting C9 incorporation and polymerization and hence obstructs all three complement activation pathways (Figure 2) [35].

Vitronectin and clusterin scavenge intermediate soluble terminal complement complexes to function as soluble inhibitors of the terminal pathway. The terminal pathway inhibitors clusterin and vitronectin block the insertion of MAC into the membrane, resulting in the formation of a soluble C5b-9 complex that can serve as a signal for complement activation [30]. Furthermore, when the complement system is activated, pro-inflammatory and chemotactic anaphylatoxins are released, potentially exacerbating inflammation, thrombosis, and vascular leakage [35]. This complement response leads to tissue damage from pro-inflammatory lesions and an increase in apoptosis. Serious disruptions in complement control may result in complications of pregnancy such as PE [36]. 

## 3. The Complement System in HIV Infection

### 3.1. Complement Activation in HIV Infection

Following complement activation, all three complement pathways promote virus opsonization and complement deposition. The complement system is essential during viral infection and serves as the first line of defence against HIV infection elimination and neutralization. When HIV-1 enters the host, it activates the complement system via glycoproteins, gp41, and gp120 without the need for HIV-specific antibodies, resulting in the virus becoming coated with complement fragments [37]. Following that, adaptive immunity is initiated, resulting in the production of anti-HIV-1 antibodies, and activated T cells [38].

The LP’s MBL protein may also interact with a variety of viral antigens, with varying effects on neutralization or viral enhancement. Mannose binding lectin has the ability to bind HIV-1 gp-120 directly. Additionally, at high doses, MBL in comparison to other complement proteins, may enhance HIV-1 pseudotyped virus infection [39]. This interaction was sufficient to prevent HIV infection of CD4+ H9 lymphoblasts in cell lines. Ying et al. [40] reported a similar finding, although much higher concentrations of MBL were required to achieve the same level of neutralization, and these findings were not replicated when infection was performed using HIV-1 primary isolates or other cell lines [40]. Mannose binding lectin was later shown to be sufficient for virus opsonization but not neutralization [18]. Complement components are deposited on HIV-1 virions downstream of MBL binding, which increases viral uptake and internalization into dendritic cells (DCs). The inhibition of C-type lectins, integrins, and CD4 reduces both complement-opsonized and complement-free HIV-1 binding. Individual blockers, conversely, revealed that complement-opsonized HIV-1 used 1- and 2-integrin for binding and uptake, whereas complement-free HIV-1 used 2- and 7-integrin [41,42].

### 3.2. Enhancement of Complement in HIV Infection

The complement cascade can proceed to the MAC’s assembly after complement deposition and opsonization. To lower the first viral load, MAC production may disrupt and lyse the lipid membranes of encapsulated viruses or kill infected cells that exhibit viral antigens. This expression of viral proteins can dysregulate and circumvent this reaction [43]. Infected host cells with viral antigens on the cell surface membrane can activate the CP since the antigens bind to IgM/IgG to trigger complement-dependent cytotoxicity, which leads to an inflammatory response. The infected cell is subsequently lysed using the MAC to minimize viral load. Complement-dependent lysis monoclonal antibodies can cross-react with H1 and H2 haemagglutinin subtypes to provide broader protection against influenza A virus infection than neutralizing monoclonal antibodies. Similarly, broadly neutralizing anti-HIV-1 antibodies bind to the viral envelope protein produced on infected primary cells to start complement deposition. The deposition has no immediate lytic effect, but it prevents viral transmission to other cells [44,45]. 

During the HIV infection budding process, HIV-1 gains complement system regulators such as CD55, CD59, and FH (Figure 3). As a result, the virus becomes complement-opsonized and is immune to complement-mediated lysis, which will increase the viral load and therefore increase infectivity [46]. Furthermore, the complement system, acting either alone or in collaboration with DCs, prime antiviral T-cell immunity, implying that the complement system promotes cell lysis and initiates an adaptive immune response [37].

After infection, cell lines exhibit varying levels of vulnerability to anti-HIV antibodies-mediated complement-dependent cytotoxicity (CDC). This may be attributed to the variable upregulation in expression of CD46, CD55, and CD59 in different cells as well as the inconsistent ability of polyclonal IgG’s to identify the envelope of the cell-associated virus. Thus, HIV-1 can escape complement-mediated lysis [47].

Antiretroviral (ART) drugs such as nucleoside- and non-nucleoside- reverse transcriptase inhibitors, respectively are protease inhibitors which are used in the treatment of HIV infection. As part of highly active antiretroviral therapy (HAART), combinations of at least three antiretroviral medications are used. This has significantly improved survival for HIV-1 infected individuals. Highly active antiretroviral therapy lowers the plasma viral load of patients and boosts CD4+ T cell counts [48], which delays the disease’s progression and lowers the risk of opportunistic infections.

## 4. The Complement System in Normal Pregnancy 

### 4.1. Immunotolerance in a Healthy Pregnancy

Pregnancy is a unique circumstance that allows the foetus to thrive and develop normally in the maternal uterus. As half the genes are received from the father, the foetus, and the placenta must be considered a ‘semi-allograft. The foetus carries paternal alloantigen’s that the mother recognizes as foreign [49]. Together, the placenta and foetus make up a genetic allograft that is fed by the mother’s blood supply. An immunosuppressed equilibrium is maintained during a healthy pregnancy. An essential component in supporting adaptive immune responses is the complement system which contributes either directly or indirectly to the development and ongoing maintenance of tolerance during pregnancy, in addition to being well-regulated on its own [19].

The placenta contains a completely functional complement system that is produced locally by a variety of cell sources and derives from the circulation of the mother. Like any other tissue in the body, the placenta’s function is to protect the foetus and the mother from pathogens and other harmful substances [50]. At the maternal–foetal interface, the placenta may be attacked by complement-mediated immunity due to the semi-allogeneic nature of foetal tissues and the frequent development of alloantibodies in the mother. This could result in foetal loss. In a successful pregnancy, the three regulatory proteins DAF, MCP, and CD59—located on the surface of trophoblasts impede unchecked complement activation [50,51]. 

Furthermore, an interaction of complement receptor 3 with iC3b at the maternal–foetal interface results in the production of local anti-inflammatory cytokine expression [52]. Furthermore, iC3b promotes the synthesis of the anti-inflammatory cytokines Interleukin 10 (IL-10) and Transforming growth factor 1 (TGF-1) in late pregnancy. Complement typically eliminates endogenous waste products with the assistance of iC3b, which is rapidly formed from C3b on host cell structures [53]. Cells with receptors for iC3b-coated particles are usually capable of generating the immunosuppressive cytokines IL-10 and TGF-1. Interleukin 10 has been shown to decrease T-cell activation by reducing the generation of proinflammatory cytokines and the upregulation of molecules involved in antigen presentation by DCs and macrophages [54]. 

Transforming growth factor beta-1 has a crucial function in blocking self-antigen-induced T-cell activation. Furthermore, C3dg and C3d deposition on antigens is an essential route of antigen absorption to antigen presentation on cells such as DCs, follicular DCs, B cells, and macrophages, as well as transport to the adaptive lymphatic system [55]. Complement inhibitors are highly expressed by cells in the placenta and the syncytiotrophoblast membrane. While placental trophoblast cells do not express classical MHC molecules, the extravillous trophoblasts (EVTs) does express tolerogenic Human leukocyte antigen (HLA-G, HLA-F, and HLA-E) molecules. The presence of HLA g on EVTs protects the placenta from cytotoxic responses [56]. In a healthy pregnancy, the invading EVTs cells contain HLA-C receptors that interact with the Killer Ig-like receptors (KIR receptors) of the uterine natural killer (NK) cells to generate an anti-inflammatory, immunotolerant maternal response to the foetus [57]. When compared to their peripheral blood counterparts, uterine NK cells have been proven to be less cytotoxic. They are required for placental development and interaction with invading EVTs cells [14].

In a healthy pregnancy, the invading EVTs cells contain HLA-C receptors that interact with the KIR receptors of the uterine NK cells to generate an anti-inflammatory, immunotolerant maternal response to the foetus [58]. Uterine NK cells are less cytotoxic than their peripheral blood counterparts. They are required for placental development and interaction with invading EVTs [59].

### 4.2. Complement Regulation in Pregnancy

Although complement activation plays a critical role in host defence, it is important to acknowledge that complement regulation and activation must be balanced carefully. Three regulatory proteins that are present on the membrane of the trophoblast cells regulate complement activation: MCP (CD46), DAF (CD55), and CD59, a glycophosphatidylinositol (GPI)-anchored protein (Figure 4). DAF promotes the degradation of preformed C3 convertase and stops the creation of new C3 convertase whereas MCP cleaves C3b and C4b into their active forms [16] (Figure 4) CD59 acts downstream to prevent the production of MAC. Complement regulation dysregulation is the cause of pregnancy illnesses including PE [60].

Mice experiments highlight how crucial complement inhibition is to a healthy pregnancy. A fundamental investigation by Xu et al. [61] discovered that 100% of neonatal death in the progeny resulted from a loss of the membrane-bound intrinsic complement inhibitory protein, murine complement receptor-1–related gene/protein Y (Crry) [61]. Nevertheless, mice devoid of both the Crry and C3 genes were fully recovered, indicating that Crry/embryos perish in utero due to an incapacity to inhibit complement activation and the ensuing complement-mediated harm [16,61]. 

## 5. The Complement System in Preeclampsia

During normal pregnancy, the maternal spiral arteries undergo a physiological conversion that allows the foetus to receive adequate blood flow. In PE, aberrant placentation, in which deficient trophoblast invasion and a lack of remodelling of the spiral arteries prevails results in a reduction in the luminal diameter of spiral arteries [62]. This results in insufficient blood supply to the foetus, failing to meet the nutrient and oxygen requirements of the foetus. This results in the placenta becoming ischaemic and releasing anti-angiogenic substances such as soluble fms-like (sFlt-1) and soluble endoglin (sEng), as well as other inflammatory mediators into the maternal circulation [63].

Complement activation at the maternal–foetal interface causes neutrophil recruitment, which leads to a rise in local tumour necrosis factor (TNF) and a decrease in vascular endothelial growth factor (VEGF), resulting in improper placentation and foetal mortality [64]. Notably, the aberrant placentation process may be reversed by inhibiting the complement pathway using inhibitors at the maternal–foetal interface. The study by Lillegard et al. [65] reported the relationship between complement activation and pregnancy hypertension, where complement inhibition slowed the evolution of high blood pressure following placental ischaemia in a rat model [65]. Furthermore, complications may arise if the complement system fails to remove ischaemic placental components [57].

A possible connection between complement dysregulation and angiogenic imbalance has been noted, and numerous studies have documented a marked increase in the prevalence of PE in women with elevated levels of sFlt-1 and sEng in the maternal circulation (Figure 5) [66]. According to a study by Banadakoppa et al. [67], sFlt-1 levels were upregulated and secreted upon complement activation in vitro on syncytial human trophoblast cells at sub-lethal levels. Additionally, the data revealed that C3a considerably raised sFlt-1 mRNA levels but not its secretion from trophoblast cells. However, sFlt-1 secretion was linked to the MAC’s release [67]. Additionally, elevated amounts of C3a were discovered, and they showed an inverse correlation with sFlt-1, indicating that sFlt-1 was exclusively elevated in trophoblast cells and was not released, staying unnoticed in maternal plasma [57].

The complement system is the first line of defence in the circulatory and tissue systems. Extravillous trophoblast cells invading maternal tissues may come into contact with complement-activating antibodies. As a result, they must be adequately protected from the maternal complement system. The syncytiotrophoblast, which is the placental surface that is always exposed to maternal blood, is a second location that requires protection from complement activation. When the complement system is activated, proinflammatory and chemotactic anaphylatoxins are released, which have the potential to cause inflammation, vascular leakage, and thrombosis [68]. Inadequately or aberrantly regulated complement activation may also cause tissue damage on the placental villi, marked by inflammatory lesions and increased apoptosis. The formation of the MAC causes calcium influx into target cells, which can generate a metabolic storm when sub-lytic C5b-9 activates apoptotic pathways and further strains the cells, resulting in unforeseen outcomes such as the haemolysis, elevated liver enzymes and low platelets (HELLP) syndrome associated with PE [69].

It has been suggested that PE is caused by an over-activation of the maternal complement system. In HDP, impaired placental perfusion is caused by over-activation of the complement system, which enhances inflammation, or furthermore, complement component deficiencies compromise the development and perfusion of the uteroplacental unit [3]. If complement control mechanisms are not adequately functioning, complement activation may occur spontaneously. Incompatibility between the maternal immune system and placental cells can lead to an imbalance in complement activation and regulation. This may play a role in the pathophysiology of PE. Complement assault may impair placental cells in discordant situations if their defence fails. Antibodies or exposed tissue structures may also act as a trigger [53].

## 6. Complement Component C1q

### 6.1. Structure & Function

The first sub-component of the CP, C1q, is an essential part of the complement system. Monocyte-lineage cells, including macrophages, immature DCs, and microglia, are known to locally synthesize C1q, albeit with minor contributions from other cell types [70]. C1q is a large (460 kDa) glycoprotein whose C-terminal is arranged into a hexameric structure made up of globular heads. This is joined to a collagen-like triple-helix tail at the *N*-terminal which holds the subunits together. Each subunit is a heterotrimer formed by three chains, C1qA, C1qB, and C1qC [71]. C1q creates the C1 complex in association with the enzymatically active elements C1r and C1s. Its targets include an array of closely spaced antigen-bound immunoglobulins IgG and IgM, C-reactive protein, and apoptotic cells. Upon binding of C1q to its targets, the enzymes C1r and C1s become active [72,73]. 

The resulting complement activation can lead to activation of the complement effector mechanisms; opsonization, anaphylatoxin release, and formation of the MAC.

### 6.2. C1q in Pregnancy 

It is accepted that C1q, a two-edged molecule, affects the course of pregnancy. C1q is widely distributed around the foetal vessels, spiral arteries, trophoblast cells, decidual endothelial cells, and EVTs. Extravillous trophoblasts invading the decidua produce C1q, which is necessary for an adequate normal placentation process as it favours the active angiogenesis in the developing decidua. It facilitates interstitial migration and serves as a molecular link between maternal endothelial cells of the spiral arteries and endovascular trophoblasts [74]. 

Previous studies demonstrated that mice lacking the C1q gene exhibited signs of PE (Figure 6), including elevated sFlt-1/placental growth factor (PIGF) ratio, hypertension, proteinuria, glomerular endotheliosis, and enhanced oxidative stress [75,76]. Based on these investigations and the previously documented function of C1q in vascular and tissue remodelling, it is plausible that pregnancy problems are related to impaired C1q production. However, Agostinis et al. [77] have also demonstrated that throughout the pregnancy, there was no apparent significant difference in C1q levels between pregnant and non-pregnant women [74,75]. Despite similar placental C1q levels, serum C1q levels were lower in PE compared to healthy pregnant women [77].

### 6.3. C1q in the Regulation of HIV Infection

Notably, since C1q binds directly to the amino acid (aa) residues 601–613 of the transmembrane protein gp41 of HIV-1, this reaction may activate the re-assembled C1 complex to a degree similar to that achieved with immune complexes (Figure 7) [78]. Furthermore, recombinant gp41 and purified HIV-1 virus attach to C1q, initiating the complement cascade [79].

Evidence indicates that the C1q-binding site is situated between residues 601 and 613 (sequence GIWGCSGKLICTT) of the immunodominant loop region of gp41 of the disulfide bridge connecting *Cys* 605 to *Cys* 611. The inhibitory activity of peptide 601–613 was significantly reduced, and its inhibitory activity was eliminated when *Ala* replaced all hydrophobic residues [79].

## 7. Complement Component Mannose-Binding Lectin

### 7.1. Structure and Function

Mannose-binding lectin is a C-type lectin belongs to the collectin family of proteins. The MBL pathway uses a protein very similar to C1q to trigger the complement activity [80]. Also, the six globular heads of MBL form a complex with two protease zymogens, which in the case of the MBL complex are MASP-1 and MASP-2. These bind to *N*-acetyl glucosamine and mannose residues, which are ubiquitous among bacteria, and form carbohydrate-recognition domains. Therefore, MBL may initiate complement activation by binding to pathogenic surfaces [13]. Notably, MBL complement activation is the same as the CP, forming a C3 convertase from C2b bound to C4b [80]. 

MBL concentration is largely inherited from parents, and its concentration is typically low in normal plasma [81]. The liver plays a crucial role in increasing MBL production during the acute-phase reaction of the innate immune response [20]. Early childhood infections are more common in those with MBL deficiencies, highlighting the importance of the MBL pathway for host defence [80]. The age range during which infections linked to MBL deficiency may occur demonstrates the significance of innate host defence mechanisms during childhood, before the child’s adaptive immune response reaches its full potential [81].

### 7.2. Preeclampsia and the MBL Pathway 

The role of MBL and pathological human pregnancies—such as PE—has not yet been completely elucidated. Studies by Than et al. [82] and similarly Celik and Ozan [83] demonstrate that significantly higher concentrations of maternal plasma and serum MBL concentration occur in PE compared to normotensive pregnant women [82,83]. Furthermore, an association between high MBL serum concentration and impaired placental perfusion was observed [82]. Additionally, data published by Sziller et al. [84] showed that PE development is prevented by the MBL codon 54 gene polymorphism, which is consistent with lowered MBL production [84]. 

Elevated complement split product accumulation may occur at the trophoblast basement membrane and within the villous stroma as well as in maternal circulation in PE. The deposition of complement components is also enhanced in the walls of uterine spiral arteries, foetal stem vessels, and kidney glomerular capillary walls [85]. Moreover, this deposition is lower in normal-term and normal-preterm placentas compared to PE [86]. According to Kitzmiller et al. [87], these findings might point to a comparable function for the MBL pathway in PE to the microvascular problems associated with diabetes, hypoxia/reperfusion injury, or kidney transplant rejection [87].

Extreme trophoblast damage would raise the risk of inadequate trophoblast invasion of spiral arteries, thereby contributing to placental hypoxia and the emergence of symptoms and traits associated with PE [88]. Additionally, maternal complement activation by MBL may contribute to the destruction of paternal antigen-expressing trophoblasts at the maternal–foetal interface. 

### 7.3. HIV Infection and the MBL Pathway 

In HIV infection, the MBL pathway plays an important role in innate immunity by binding to carbohydrates on the surface of microorganisms, including HIV-1 gp120/gp41 [89]. The extensively glycosylated envelope protein (gp120/gp41) of HIV-1 offers a potential target for an innate immune system attack via the C-type MBL. Numerous investigations have unequivocally shown that this binding of MBL to HIV-1 causes complement activation [90,91] (Figure 8). 

Of note, the high mannose glycans on gp120 are required for the attachment of MBL to HIV-1. Notably, it has been demonstrated that the MBL pathway interacts with every studied strain of HIV-1 because it is selective for the glycan types that are prevalent on gp120. Although evidence of direct neutralization of HIV infection generated in T cell lines by MBL exists, neutralization for HIV infection distinctive isolates is comparatively low. Nonetheless, medications that change how carbohydrates are processed improve MBL’s ability to neutralize HIV-1 main isolates [92]. Moreover, complement activation on gp120 and HIV opsonization as a result of MBL binding have been noted. HIV binding and opsonization by MBL may change HIV infection-related viral trafficking and viral–antigen presentation [91]. Increased MBL levels correlate with disease progression in people living with HIV. The complement activation capacity through the MBL pathway is boosted in tandem with the increase in MBL, indicating greater MBL-mediated complement activation during HIV infection progression [93].

## 8. Complement Component C2 

### 8.1. Structure and Function

C2 is an essential part of the CP and LP that prevents microbial infections from occurring and eradicates immune complexes. MBL or ficolin together with MASP-1 adhere to carbohydrate molecules. By cleaving C2 and C4, this activates MASP-2 and produces a C3 convertase that resembles the CP [94]. Chromosome 6 has a short arm, known as HLA class III, that contains complement component C2. This precursor protein is created when C1 is activated to produce the C2b and C2a components. Despite similar sequences of C2 and serine proteinase, the former has a catalytic chain with an expanded *N*-terminus containing 60 amino acids [95].

Also, the fundamental structure of Factor B and C2 are comparable. Together with C4b, C2a generates C3 convertase (C4b2a). 

### 8.2. C2 in HIV Infection and Preeclampsia 

The exact function of C2 in pregnancy is unknown, however, a study by Johnson and Gustavii [96] found that the maternal blood of normal pregnant women had higher concentrations of complement proteins (C2, C4, C3, C5, C6, and Factors B and H) than non-pregnant women [96]. On the other hand, disruption of the complement process may result in the development of autoimmunity, and some chronic inflammatory diseases are also linked to C2 polymorphisms and deficiencies: both systemic lupus erythematosus (SLE) and age-related macular degeneration [97]. The rate of complement activation may be independently influenced by the functional differences amongst C2 variants. The LP and CP are less activated in C2 deficiency. Increased levels of immune complex in these individuals make them more vulnerable to autoimmune disorders. There is a 10% penetrance of C2 deficiency that occurs in PE, as well as SLE since they have a comparable aggravated immunological milieu. Furthermore, complement-mediated injury occurs in pregnant SLE women, making them more susceptible to miscarriage, placental insufficiency, preeclamptic development, and foetal growth restriction [19,98].

Notably, little is known about C2 in HIV infection. Combining complement activation with antibody activation can render HIV inactive. HIV infection neutralization progresses via the CP, as demonstrated by the requirement for an antibody response and the observation that C2-deficient serum plus antibodies do not release reverse transcriptase. This neutralizing process is comparable to what has been demonstrated for numerous other enveloped viruses [99]. Huson et al. [100] found that asymptomatic HIV infected patients had higher levels of C3 and C1q-C4 compared to healthy controls. Nevertheless, a decrease in MBL does not affect complement activation, indicating that HIV infection largely activates the complement system through the CP [101].

## 9. Complement Component C3

### 9.1. Structure and Function

Complement component C3 is the central component of the complement system and was first discovered in 1960 by Müller-Eberhard and his associates [102]. C3 is integrated as a 1663 amino acid pre-pro protein, and when it enters the complement pathways, the 22-residue signal peptide found on its surface is removed, resulting in the formation of a 1641 residue pro-protein. This new Pro-C3 protein complex is oxidized during the folding process, creates 13 pairs of disulfide bonds, and becomes post-translationally changed at three different locations (Asn-85, Asn-939, and Asn-1617). Preceding to export from the cell, the furin proteases cleave pro-C3 into two distinct polypeptides known as the β-chain (residues 23–667; ~75 kDa) and α-chain (residues 672–1663; ~110 kDa), which remain covalently bonded to one another via an intermolecular disulfide bond in native C3 (~185 kDa) [103].

C3 is mostly expressed by the liver and circulates in plasma; however, most human cell types express some level of C3. C3 is considered biologically inactive, but its cleaved fragments (C3a, C3b, iC3b, C3d, and C3d) have numerous biological functions [104]. C3 activation fragments act as inflammatory modulators (anaphylatoxin C3a) or opsonin’s (C3b, iC3b, and C3d) with context- and/or receptor-dependent pro- and anti-inflammatory, destructive, or protective activities [103]. C3 and its cleaved fragments serve multiple functions throughout the complement cascade such as initiation of complement activation (Figure 9) via tick-over of C3 and passive adsorption to surfaces, as well as the potential surface capturing of C3b by modulators such as properdin or P-selectin; as a substrate (C3) and C3 convertase component (C3b); and as a direct effector protein to mediate phagocytosis and immune modulation (C3a, C3b, iC3b, C3dg) [105].

### 9.2. C3 in Preeclampsia and HIV Infection

Complement C3 levels are upregulated in both disorders. The most likely explanation for this increase in C3 is its cleavage into C3a in inflammatory conditions such as PE and HIV infection. C3a is a strong anaphylatoxin that increases vascular permeability and smooth muscle contraction, most likely as a compensatory response to the absence of spiral artery transformation in PE; and endothelial damage occurring from HIV infection and PE [106]. Additionally, C3a is a leukocyte chemotactic protein that may activate neutrophils and monocytes in a preeclamptic environment, promoting the production of inflammatory mediators such as free oxygen radicals, proteases, and pro-inflammatory cytokines, contributing to the hyperinflammatory state and excessive complement activation observed in PE [107]. Elevation of C3 occurs in both PE and normal pregnancy, from the start of the pregnancy until parturition [108]. He et al. [7], reported that the concentration of C3 elevated in the first trimester and continued throughout the second semester but decreased in the third trimester; this may be part of the acute phase reaction in the third trimester of both normotensive and preeclamptic pregnancies [7]. The study by Kennelly et al. [109] reported similar findings: this upregulation in C3 is a result of aberrant complement activation, induced by PE [109].

HIV infection causes a chronic inflammatory state due to complement activation via envelope proteins on the HIV virion [110]. In HIV infection, C3 the vital component of the complement system becomes cleaved into C3a, a potent anaphylatoxin that releases pro-inflammatory cytokines and chemical mediators, thus creating a hyperinflammatory microenvironment and causing widespread endothelial damage [104].

## 10. Complement Component C4

### 10.1. Structure and Function

Complement C4 is a descendant of the same ancestral gene as complement proteins C3 and C5 and shares up to 30% sequence identity with these two complement proteins [111]. However, C4 maturation is more complex compared to that of C3 and C5, which includes the generation of three chains; α (95 kDa), β (75 kDa), and γ (30 kDa), from the precursor and the introduction of posttranslational modifications, including four *N*- and one *O*-linked glycosylation and sulfation of three tyrosine residues [112]. C4 has two isotypes: those produced by the two genes C4A and C4B. Although C4A and C4B differ by just four amino acid residues (C4A: P_1120_PCPVLD_1125_; C4B: L_1120_LSPVIH_1125_), they have drastically differing haemolytic activity and substrate affinities [113]. Complement C4 is essential for the activation of CP and LP, as well as the synthesis of C3 convertase, which results in the development of the MAC against pathogens and other foreign substances [113].

### 10.2. Preeclampsia and HIV Infection

In PE, the complement system is aberrantly regulated, leading to excessive amounts of circulating complement activation products in both early-onset PE and late-onset PE [57]. C4 is cleaved into its fragment, C4d, which is found to be deposited in excess on the placenta of PE compared to normal pregnancies. Additionally, C4d is also observed in the endothelium of kidney glomerular capillaries of PE. Extravillous trophoblast cells and first-trimester cytotrophoblasts express excess complement C4, in an IFN-γ driven process. These findings are indicative of widespread endothelial damage and an exacerbated inflammatory response of PE. Furthermore, these findings are corroborated by the study conducted by Lokki et al. [14], which reported C4 gene products, C4A and C4B to be correlated with the severity of PE [14]. 

The study conducted by Liu et al. [114], reported an upregulation of complement C4 in HIV-infected individuals with neurocognitive disorders [114]. Similar findings were also observed by Talagrand-Reboul et al. [110]. These findings are indicative of excessive complement activation, stemming from an induced immunological response in HIV infection [110]. During HIV infection, complement C4 is cleaved into C4a and C4b by the CP and the MBL pathway, as an immunological reflex action [100]. Furthermore, HIV infection reduces protein S levels, which have been linked to the attachment of C4 binding protein, a key regulator of the complement system, to the surface of dying cells. As a result, in the presence of protein S shortage, apoptotic cells may initiate uncontrolled C4 activation [101].

## 11. Complement Component C5 

### 11.1. Structure & Function

Complement component 5 plays an important role in inflammatory and cell-killing processes. This protein is composed of alpha and beta polypeptide chains that are linked by a disulfide bridge. Moreover, the activation of the complement system via the complement pathways may lead to the increased synthesis of C3 and C5 convertases, resulting in the cleavage of C5 into C5a and C5b. C5a plays an important role in chemotaxis [115,116]. C5b forms the first part of the complement MAC [116]. 

### 11.2. C5 and Preeclampsia 

The main manifestation of dysregulated complement activation in PE is high expression of C3a, C5a, and MAC compared to normal pregnancy [6,16,62]. 

Additionally, soluble vascular endothelial growth factor receptor-1 (sVEGFR-1), commonly referred to as sFlt-1, is a strong anti-angiogenic factor that may be released by C5a [117]. Pro-angiogenic VEGF and PIGF are antagonistic to sFlt-1, ensuring a healthy pregnancy. According to Girardi et al. [49], there is an increase in circulating levels of sFlt-1 in both murine models of pregnancy loss and women who experience recurrent miscarriages [88] Several investigations provide compelling evidence of increased levels of circulating sFlt-1, together with a concurrent decrease in PIGF and VEGF, in women experiencing recurrent pregnancy loss or developing PE (Figure 10) [16,118,119].

### 11.3. Therapeutic Evidence of C5 in Preeclampsia 

The concept of complement-targeted therapy is intriguing because complement activation has been demonstrated. The US Federal Drug Administration has approved the drug Eculizumab for the treatment of anomalies related to cryopyrin-associated autoinflammatory syndromes, paroxysmal nocturnal haemoglobinuria (PNH), and refractory myasthenia gravis [120]. It is a recombinant humanized IgG2/IgG4 kappa monoclonal antibody that binds to complement component C5 to specifically target and block the terminal phase of the complement cascade [121]. In vitro, evidence of complement dysregulation in PE, HELLP syndrome were used to illustrate the effectiveness of Eculizumab in treating PE in comparison to normal and non-pregnant controls. Further supporting the complement dysregulation in both disorders, the same study also found lower in vitro cell killing when Eculizumab was added to HELLP serum [120]. According to Kelly et al. [122], there have been numerous cases of Eculizumab being used throughout pregnancy for atypical haemolytic uremic syndrome and paroxysmal nocturnal haemoglobinuria without a discernible negative impact on the mother or the foetus, indicating that the medication is safe to use during this delicate period [122].

### 11.4. C5 and HIV Infection

Tumour-necrosis factor-α and IL-6 secretion from monocytes and monocyte-derived macrophages may be enhanced by the complement activation product C5a, an anaphylatoxin [123]. Moreover, DCs stimulate C5a, which is brought on by HIV-1 and helps autologous primary T cells proliferate. Apart from augmenting HIV-1 infectivity through complement activation, HIV-1 also potently stimulates the manufacture of complement factor 3 in neurons and astrocytes [124]. The pure HIV-1 viral proteins Nef and gp41 are biologically active in upregulating C3, but Tat, gp120, and gp160 are unable to influence C3 production [123].

## 12. Complement Component C9 and Membrane Attack Complex (MAC)

### 12.1. Structure and Function

C9 is a membrane protein that functions as a sentinel host defence against pathogenic assault. It also has a role in the aetiology of certain liver illnesses, as well as liver injury and healing [125,126]. Complement component 9 of MAC is involved in pore creation. 

The MAC is formed by the successive assembly of the soluble complement proteins C5b, C6, C7, C8, and C9. Membrane attack complex can be assembled experimentally using heterologous sera4; zymosan-activated sera, which results in complement activation in the fluid phase via the AP and generates soluble MAC14 in the absence of cell membranes; sequential addition of recombinant C5b6, C7, C8, and C9; and high-titre panel reactive antibodies, which react primarily with non-self-class I and class II major histocompatibility complex molecules on endothelial cells. It functions to target bacteria as well as encapsulate parasites and viruses [127]. C9 can also create poly-c9 structures that are similar to the MAC pore. Several copies of C9 are recruited in sequence to membrane associated C5b8 during MAC formation to form a pore and hence trigger lysis [128].

### 12.2. C9 and MAC in Preeclampsia and HIV Infection

The MAC also triggers a variety of physiologic changes, including apoptosis and the release of pro-inflammatory cytokines [129]. The release of pro-inflammatory cytokines (TNF-α, IL-2) induces excessive inflammation, which leads to increased complement activation, as seen in both HIV infection and PE [3]. Brewster et al. [130] discovered elevated levels of the C5b-9 MAC in PE, indicating excessive complement activation (Figure 11) and hyperinflammation. Sun et al. [131] discovered that complement signalling pathways were downregulated in HIV-co-infected patients [130,131]. This is most likely owing to the immune system weakening effects of HIV infection [132]. Excessive production and release of physiologically active products during complement cascade activation caused by PE and HIV infection may result in tissue injury via the MAC [11].

## 13. The Complement System in HIV-Associated Preeclampsia 

Different complement components are increased in PE, indicating an elevation in complement activation (Figure 12), which can lead to opsonization and an increase in the number of HIV virions in HIV-infected women [37]. In PE, an increase in inflammatory cytokines by pro-inflammatory T cells and a decrease in anti-inflammatory and regulatory cytokines generate an imbalance that leads to persistent immune activation [133]. 

With the use of HAART, HIV infection causes comparable chronic immunological activation. Excessive or abnormal complement activation might result in an inflammatory milieu. Excessive or dysregulated complement activation recruit’s leukocytes, which release pro-inflammatory cytokines and anti-angiogenic mediators, inducing an inflammatory state as observed in PE [134].

Complement activation dysregulation can shift the balance from a normal to a hyper-inflammatory state. C3a and C5a are complement components that are upregulated in PE and can cause inflammation as well as recruit macrophages and DCs for HIV infection (Derzsy et al., 2010 [6]). In the presence of C5a, pro-inflammatory cytokines such as IL-6 and TNF-α increased, promoting HIV-1 infection. As a result, the chronic inflammatory state seen in HIV infection and PE may be exacerbated further by excessive complement activation [15].

## 14. Conclusions

This review has discussed the components of the complement system and their activity in PE and HIV infection. During pregnancy, the complement system is the initial barrier to our adaptive immune response, contributing directly or indirectly to the formation and maintenance of foetal immunotolerance. As a result, in the combination of PE and HIV infection, homeostatic complement activation and modulation are critical to ensuring pregnancy success. Excessive complement activation leads to upregulation of pro-inflammatory cytokines and chemical mediators, resulting in a hyperinflammatory state. Furthermore, in the combination of HIV-associated PE, the dysregulation of complement activity within this enhanced inflammatory milieu enhances endothelial cell damage, which is common of both diseases. To fully understand complement activation and regulation in HIV-associated PE, further research is required. Additionally, the complement components may function as biomarkers, which could ultimately result in more effective treatment methods through tailored drug therapy.

## Figures and Tables

**Figure 1 ijms-25-06232-f001:**
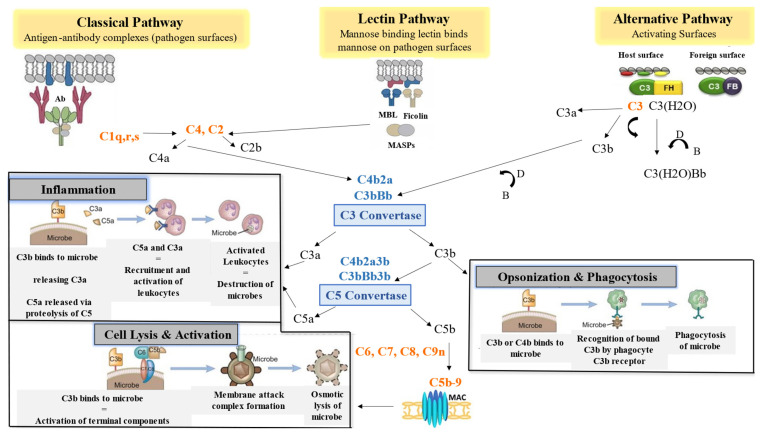
Schematic diagram showing the activation of the complement cascade. There are three pathways via which complement is activated: CP, LP, and AP. The AP is activated when C3 spontaneously hydrolyses at low concentrations to create C3(H2O), the LP is activated when ficolin or MBL attaches to carbohydrate moieties on pathogen surfaces, and the CP is triggered by antibody attachment to cell surfaces that expose a C1q binding site. All three routes combine to generate a C3 convertase, which then cleaves C3a and C3b to cause opsonization, inflammation, and the MAC, promoting cell lysis.

**Figure 2 ijms-25-06232-f002:**
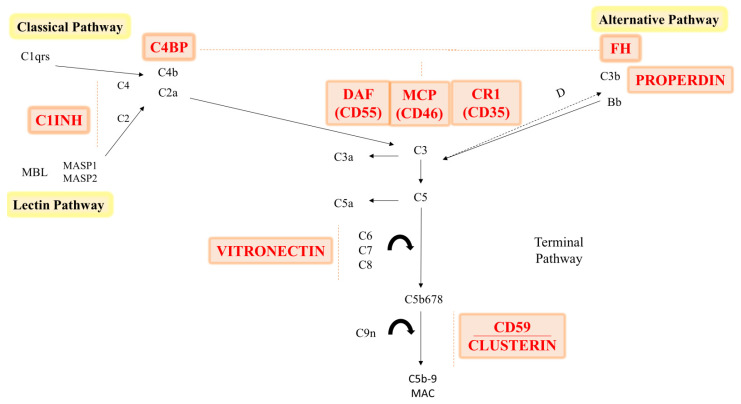
Schematic diagram showing key complement regulators and their targets in the complement system.

**Figure 3 ijms-25-06232-f003:**
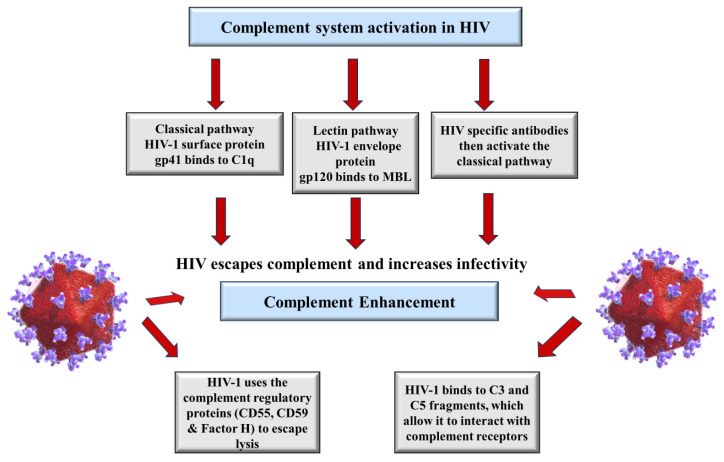
Schematic diagram of the role of the complement system during HIV infection. It is activated by HIV-1 virus binding to gp41 and gp120. HIV-specific antibodies also contribute to complement activation. HIV-1 uses the regulatory proteins to evade complement-mediated lysis.

**Figure 4 ijms-25-06232-f004:**
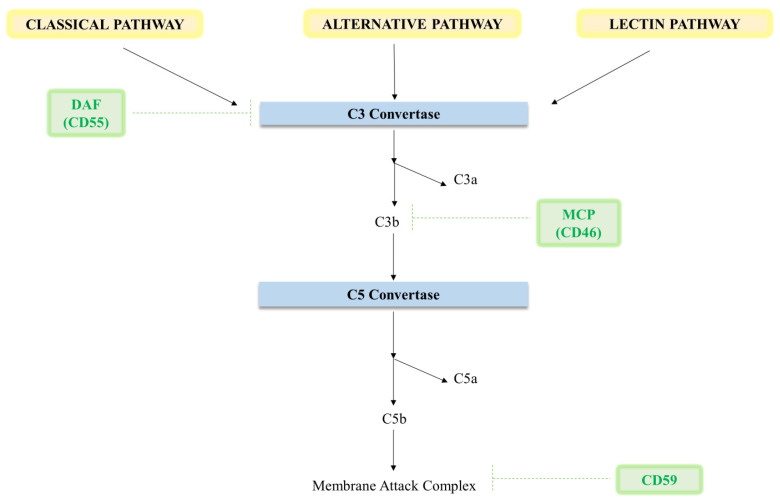
Schematic diagram showing the overview of the main effectors and regulators of the complement system during pregnancy. The primary complement regulators on human placental tissue, MCP, and CD59, are responsible for preventing inappropriate complement activation.

**Figure 5 ijms-25-06232-f005:**
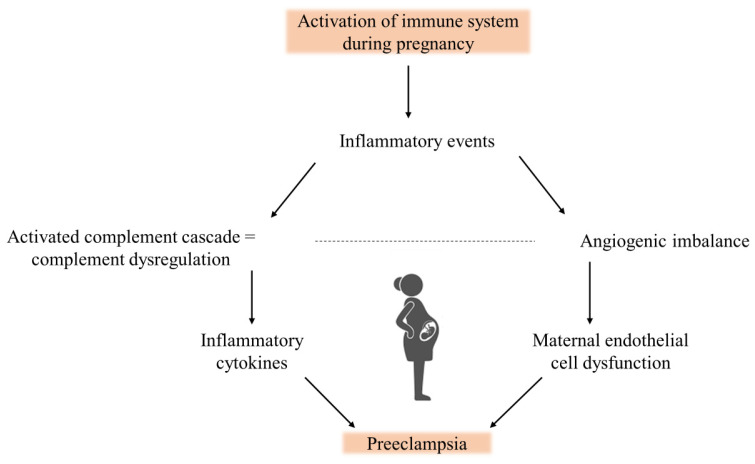
Preeclampsia in early pregnancy is caused by immune activation, specifically by variables associated with angiogenesis and complement activation fragments.

**Figure 6 ijms-25-06232-f006:**
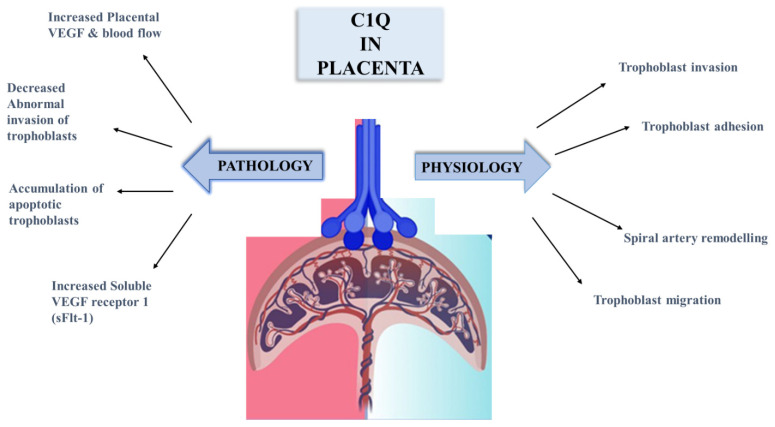
The function of C1q in normal placentation and adverse pregnancy outcomes.

**Figure 7 ijms-25-06232-f007:**
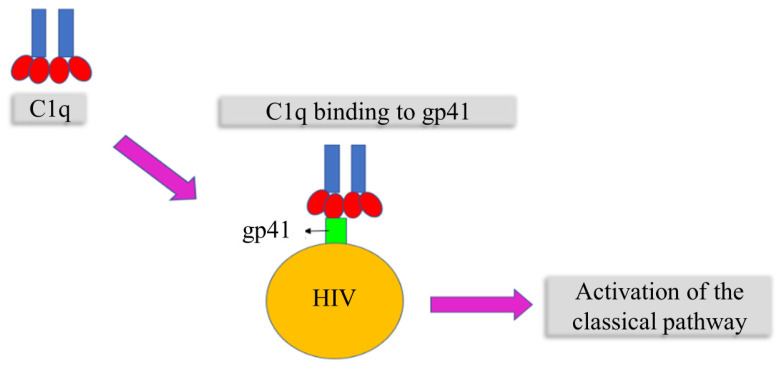
Complement activation by HIV, adhesion of C1q, and transmembrane protein gp41 leading to the CP being activated.

**Figure 8 ijms-25-06232-f008:**
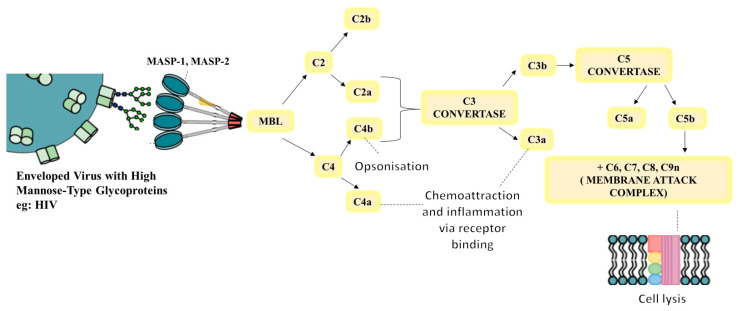
The complement activation LP. When MBL interacts with viral glycoproteins through glycan-associated mannose, conformational changes occur in MBL. This triggers MASP-1 and MASP-2, which cleave complement components resulting in inflammation, pathogen, and infected cell lysis, opsonization, and inflammation.

**Figure 9 ijms-25-06232-f009:**
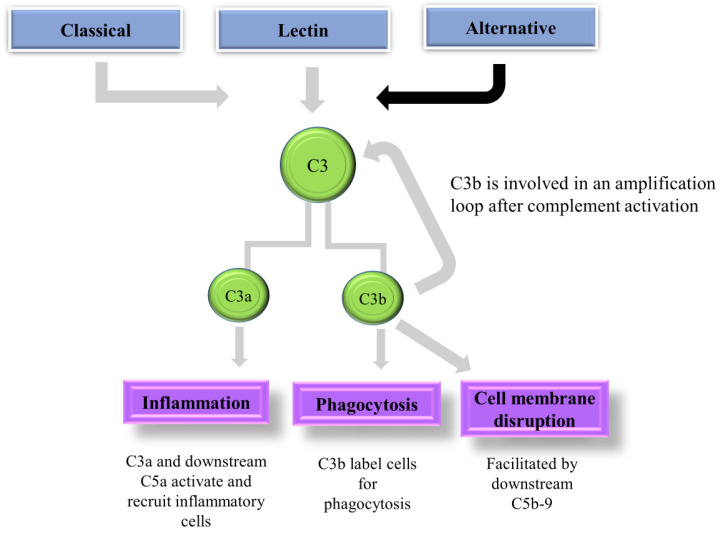
Schematic illustration of complement C3 and its function. All three complement pathways form C3, which is cleaved into C3a and C3b. C3a activates and recruits inflammatory cells, whilst C3b induces phagocytosis and cell membrane disruption.

**Figure 10 ijms-25-06232-f010:**
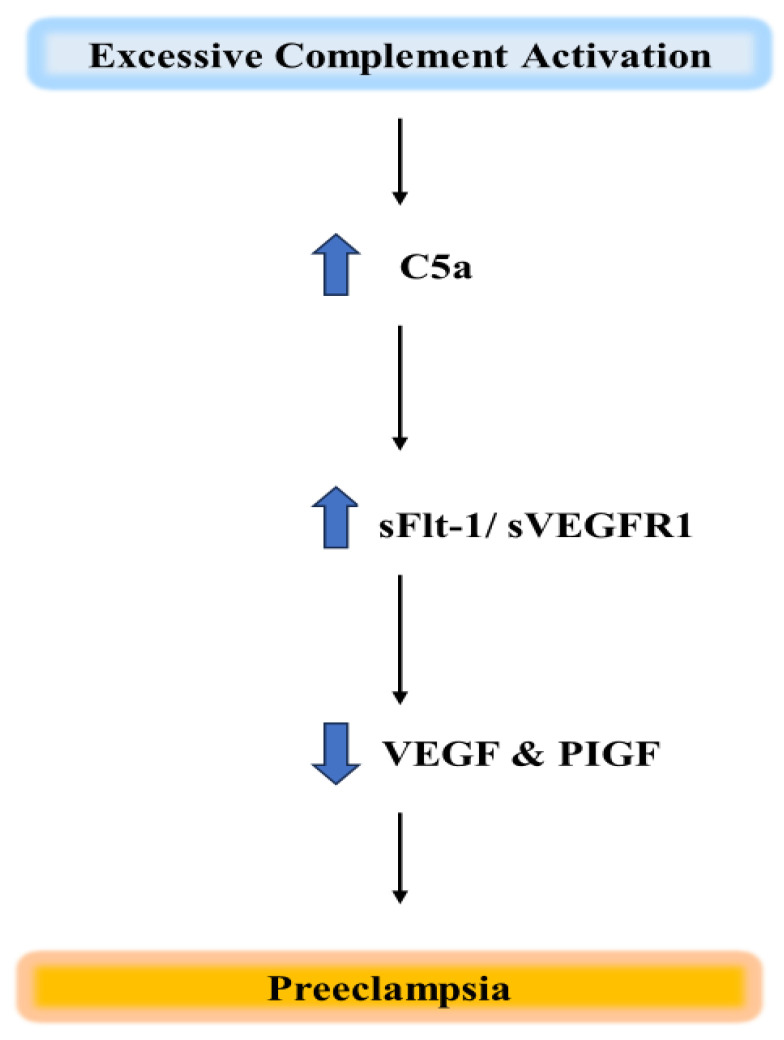
Diagrammatic representation of C5a’s function in preeclamptic pregnancies. Unfavourable excessive complement activation accompanied by an increase in C5a release increased soluble VEGF receptor-1 (sVEGFR-1)/sFlt-1. Reduced VEGF and PIGF levels cause placental insufficiency resulting in PE in late pregnancy.

**Figure 11 ijms-25-06232-f011:**
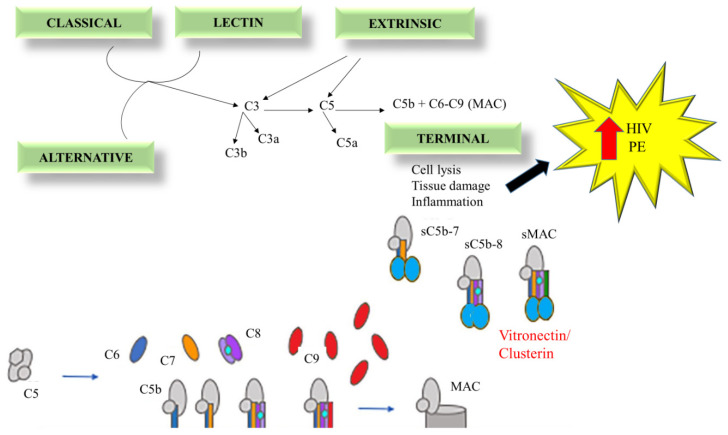
Schematic overview of the MAC in the synergy of HIV infection and PE. All three complement pathways form C5, which cleaves into C5a and C5b, which then forms the MAC by recruiting the other complement components (C6–C9). Thus, increasing complement activation in HIV and PE.

**Figure 12 ijms-25-06232-f012:**
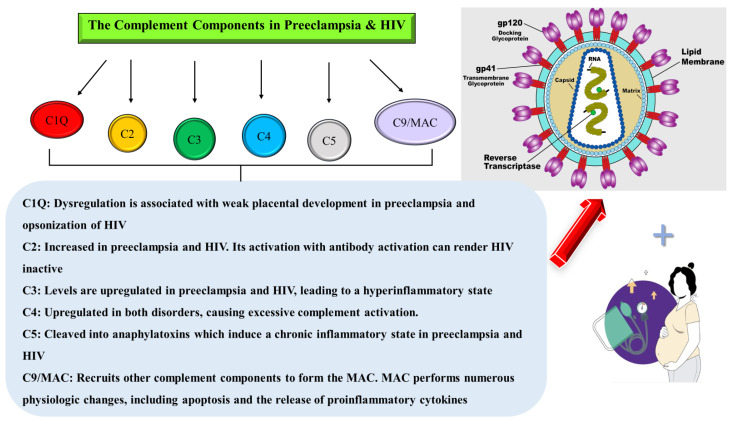
Schematic overview showing the main functions of each complement component in the duality of PE and HIV infection.

## Data Availability

Not applicable.

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
