# Peer review of "Is the Complement System Dysregulated in Preeclampsia Comorbid with HIV Infection?"

_ijms, 2024, doi:10.3390/ijms25116232_

Round 1

Reviewer 1 Report

Comments and Suggestions for Authors

Review of ijms-3017630

Is the complement system dysregulated in preeclampsia comorbid with HIV infection?

The complement system is a key component of the innate immune system that plays a vital role in host defense and homeostasis. In this review, it is described the role of the complement system in the pathogenesis of HIV infection and preeclampsia (PE), both of which represent major causes of maternal death in South Africa.

Abstract

Line 15. Delete “elusive, but it is believed to be”

Lines 18-19. Clarify this sentence “It can protect the host against HIV infection and enhance HIV infectivity”. 

Lines 19-21. Pease clarify “An upregulation of regulatory proteins has been implicated as an adaptive phenomenon in response to elevated complement-mediated cell lysis in HIV infection, further aggravated by preeclamptic complement activation”.

Comment: The authors need the clarify what happens first during viral load clearance, and how the inflammation processes as well as the upregulation of regulatory proteins are involved”

Comment: The abstract section needs some edition, for instance, the authors need to clarify the relationship between PE and HIV infection as PE represents an exaggerated immune response, while HIV infection is associated with a decline in immune activity.

What is the prevalence of PE mortality and morbidity?, What is the prevalence of PE and HIV ?

Introduction

Line 47. Please add a paragraph describing how the pathophysiology is related to inflammatory disorders?, Is this related only to the PE activity?.

Lines 51-59.  Please  edit the paragraph  “The synergy of HIV infection and PE comes from antagonistic immune responses, as PE is associated with an exaggerated immune response, whilst HIV infection dampens the immune response (Clouse et al., 2020). Notwithstanding, adaptive and innate immunity are connected through the complement system (Carroll, 2004). The complement system forms part of our innate immune response and functions to opsonize target surfaces, induce pro-inflammatory responses, and lyse cells and pathogens. Furthermore, it defends the host by removing apoptotic cells, damaged tissue, and immune complexes, ensuring homeostasis maintenance (Lokki et al., 2014).”

Comment: The authors are subsequently describing in detail the complement system, but they need to link their connection in lines 51-59 to show the importance of this interplay to support their hypothesis. There are important missing information in this section,

Line 74. Delete “by the release of antibodies”

Lines 74-75. Use only one terminology “ by natural antibodies or upon a humoral response”. Both are the same.

Line 160. Delete “may also”

Line 175. Delete “.”

Lines 177-178, Edit the paragraph  as in its current way it is hard to follow “This response is largely determined by the infectious agent, suppresses viral infection, is dysregulated by viral protein expression and may enhance viral infection”

Lines 183-184. .“Following that adaptive immunity is initiated, resulting in the production of anti-HIV-1 antibodies, and activated T cells”.

Question: During this point of adaptative response, viral load increases?, infection progresses?, or HIV infection is eliminated? , please discuss it.

Lines 195. When virus is opsonized but not neutralized, means that infection can occur?, please discuss in the review

Lines 770-772. Delete “2. Results. This section may be divided by subheadings. It should provide a concise and precise description of the experimental results, their interpretation, as well as the experimental conclusions that can be drawn.

Line 342-344. Please delete or edit the paragraph “There has been evidence of elevated C5a expression in placental trophoblasts of PE mothers and increased C5a deposition in macrophages. It was discovered that trophoblast migration and tube formation were inhibited by C5a, but that these processes were restored by C5aR knockdown (Ma et al., 2018)” Because in its current way they are hard to follow.

Lines 399-400. The paragraph can be deleted as it has been mentioned before “Thus. he removal of immune complexes, pathogens, and apoptotic bodies from the body is C1q's primary physiological activity (van de Bovenkamp et al., 2021)”.

Line 435. Edit the sentence “This binding suggests that gp41 precipitates activation of the CP without the need of Antibodies”. Delete “suggest”, This has been reported before.

Linea 454-463. This paragraph can be edited as in the current way it is hard to follow. Please summarize this section to avoid rewording “The majority of the genes that affect an individual's blood, MBL concentration are  inherited from their parents (Nørgaard-Pedersen et al., 2022). The concentration of MBLis low in the normal plasma of most individuals. The liver enables increased production during the acute-phase reaction of the innate immune response (Walport, 2001). Early childhood infections are significantly more common in those with MBL deficiencies, suggesting the significance of the MBL pathway for host defence (Dobó et al., 2016). The age range during which infections linked to MBL deficiency may occur demonstrates the significance of innate host defense mechanisms during childhood. This is the period before the child's adaptive immune response reaches its full potential and after the loss of maternal antibodies that are transferred via the placenta and colostrum (Yongqing et al., 2012).  

Line 775. The conclusion section must be improved, the authors must describe in this section the major outcome of the study. In PE and HIV, the complement is crucial, but in its current way the conclusion section is merely an extension of the review.

This review is a challenge,  Several complement components are elevated in PE which could further enhance HIV infectivity, whereas complement opsonized HIV-1 can modulate DC response and their cross-talk with NK cells to promote the upregulation of factors associated with immune suppression possibly attenuating the constant inflammatory state of PE.

What can we assume from that? Neutralization of the Immune Response can occur?, how?,  Also, the authors have described the major complement components, can the authors are able to identify which could serve as putative biomarkers?.

Comments on the Quality of English Language

English edition is needed, the document is quite long authors need to summarize some sections 

Reviewer 2 Report

Comments and Suggestions for Authors

Dear editors, thank you very much for the opportunity to review this article.
South Africa bears a huge burden in the fight against the HIV epidemic.
The work clearly explains the intricacies of pregnancy development in HIV patients and the course of the disease in this special group.

The authors constructed the article outside the suggested pattern, but by accident, I think, they left the paragraph about the results on page 19.

Round 2

Reviewer 1 Report

Comments and Suggestions for Authors

Dear authors, I am quite concern because in this second version of the Ms. None of my queries were addressed. In the responsive letter, the authors replied some of my comments, but no changes were observed in the submitted Ms.  I am dissapointed for this situation.

Comments on the Quality of English Language

English needs edition

Author Response

Dear Reviewer, thank you for taking the time to review the manuscript and send through the corrections, I truly do appreciate it. 

I have in fact addressed all the comments made by you to the best of my ability. The corrections were changed and highlighted in red writing (mentioning each line) in the new manuscript that was uploaded. Perhaps there was an issue with the upload,  but I will send it through again.

once again, I thank you for your revisions.

Round 3

Reviewer 1 Report

Comments and Suggestions for Authors

Thank you for addressing all my queries.

Comments on the Quality of English Language

Some english edition is needed